# Minimally Invasive Treatment of Stress Urinary Incontinence in Women: A Prospective Comparative Analysis between Bulking Agent and Single-Incision Sling

**DOI:** 10.3390/healthcare12070751

**Published:** 2024-03-29

**Authors:** Lorenzo Campanella, Gianluca Gabrielli, Erika Chiodo, Vitaliana Stefanachi, Ermelinda Pennacchini, Debora Grilli, Giovanni Grossi, Pietro Cignini, Andrea Morciano, Marzio Angelo Zullo, Pierluigi Palazzetti, Carlo Rappa, Marco Calcagno, Vincenzo Spina, Mauro Cervigni, Michele Carlo Schiavi

**Affiliations:** 1Department of Obstetrics and Gynaecology, Ospedale Sandro Pertini, 00157 Rome, Italy; gianluca.gabrielli@aslroma2.it (G.G.); erika.chiodo@aslroma2.it (E.C.); vitaliana.stefanachi@aslroma2.it (V.S.); ermelinda.pennacchini@aslroma2.it (E.P.); debora.grilli@aslroma2.it (D.G.); giovanni.grossi@aslroma2.it (G.G.); pietro.cignini@aslroma2.it (P.C.); pierluigi.palazzetti@aslroma2.it (P.P.); michele.schiavi@aslroma2.it (M.C.S.); 2Department of Obstetrics and Gynaecology, Università di Tor Vergata, 00133 Rome, Italy; 3Department of Obstetrics and Gynaecology, Pia Fondazione Cardinale G. Panico, 73039 Tricase, Italy; drmorciano@gmail.com; 4AIUG Research Groups, Associazione Italiana di UroGinecologia e del Pavimento Pelvico, 00168 Rome, Italy; m.zullo@unicampus.it (M.A.Z.); carlodok@gmail.com (C.R.); info@maurocervigni.it (M.C.); 5Department of Week-Surgery, Policlinico Universitario Campus Bio Medico, 00128 Rome, Italy; 6Andrea Grimaldi Medical Care, 80122 Naples, Italy; 7Department of Obstetrics and Gynecology, Santo Spirito Hospital, 00193 Rome, Italy; marco.calcagno@aslroma1.it; 8Maternal and Child Department, S. Camillo de Lellis Hospital, 02100 Rieti, Italy; v.spina@asl.rieti.it; 9Department of Female Pelvic Medicine and Reconstructive Surgery, Istituto Marco Pasquali ICOT, 04100 Latina, Italy

**Keywords:** stress urinary incontinence, single-incision sling, urethral bulking agents, Intrinsic Sphincteric Deficiency, midurethral sling

## Abstract

Introduction: The study aims to compare the efficacy and safety of bulking agents and single-incision slings in the treatment of urinary incontinence in 159 patients during a 29-month follow-up period. Material and methods: Of the 159 patients suffering from stress urinary incontinence, 64 were treated with bulking agents (PAHG Bulkamid^®^) and 75 with a single-incision sling (Altis^®^). The ICIQ-UI-SF (Incontinence Questionnaire-Urine Incontinence-Short Form), PISQ-12 (Pelvic Organ Prolapse/Urinary Incontinence Sexual Questionnaires short form), FSFI (Female Sexual Function Index), FSDS (Female Sexual Distress Scale), and PGI-I (Patient Global Improvement Index) were used to assess efficiency and quality of life. Results: The bulking agents showed high efficacy and safety during the 29-month follow-up. Post-operative complications were recorded in both groups, with only two significant differences. The Bulkamid group experienced no pain, while 10.8% of the ALTIS group experienced groin pain and 5% experienced de novo urgency. Furthermore, patients treated with bulking agents experienced reduced nicturia (0.78 vs. 0.92 in patients treated with single-incision slings.). In both groups, we noticed a significant improvement in QoL (quality of life), with a halved ICIQ-UI-SF (International Consultation on Incontinence Questionnaire-Urine Incontinence-Short Form) score which was completed to assess the impact of urine symptoms. After 24 months of therapy, the Bulkamid group saw a decrease from 14.58 ± 5.11 at baseline to 5.67 ± 1.90 (*p* < 0.0001), whereas the ALTIS group experience a decrease from 13.75 ± 5.89 to 5.83 ± 1.78. Similarly, we observed an improvement in sexual function, with the number of sexually active patients increasing from 29 to 44 (56.4%) in the Bulkamid group (*p* = 0.041) and from 31 to 51 (61.7%) in the ALTIS group (*p* = 0.034). According to the most recent statistics, the PISQ-12, FSFI, and FSDS scores all demonstrated an improvement in women’s sexual function. Conclusions: In terms of efficacy and safety, bulking agents had notable results over the 29-month follow-up period. Furthermore, the patients treated with bulking agents reported a lower incidence of postoperative complications and a no discernible difference in terms of quality of life and sexual activity compared to the ones treated with single-incision slings. Bulking agents can be considered a very reliable therapeutic option based on accurate patient selection.

## 1. Introduction

The prevalence of urinary incontinence ranges from 10 to 60% of non-pregnant women above 20 years of age and from 50 to 70% of women older than 60 years of age [1,2,3,4].

In women with stress urinary incontinence (SUI), involuntary urine leakage occurs when intra-abdominal pressure increases (e.g., with exercise, sneezing, coughing, or laughing) in the absence of a bladder contraction. [5].

SUI is hypothesized to be caused by a lack of mechanical support of the urethra and/or poor coaptation of the urethral tissues, resulting in insufficient resistance to urine outflow with elevated abdominal pressures.

The two main mechanisms involved in the physiopathology of SUI are urethral hypermobility, which develops when pelvic-floor muscles and vaginal connective tissues provide insufficient urethral support, and Intrinsic Sphincteric Deficiency (ISD), which is caused by a loss of intrinsic urethral mucosal and muscle tone.

The usual argument for urethra support playing an important role in stress incontinence is the fact that urethral support operations are able to treat stress incontinence without changing urethral function [6].

However, contrary to the “pelvic-centric” theory, a new “urethro-centric” hypothesis is emerging, according to which urethral hypermobility is a characteristic that can be both associated and not with the condition of ISD but does not constitute the etiology of SUI.

This idea emphasizes how urethral hypermobility is not the primary cause of SUI and that ISD plays a critical part in the progression of this pathologic process [7,8].

The debate on the predominance of either of these two causes has dominated the urogynecological scene in terms of the interpretation of SUI physiopathology.

Worldwide, there is an ongoing search for increasingly minimally invasive urinary incontinence surgeries. In England, there was a 30% decrease in the use of urogynecolgical surgical mesh for SUI following the FDA warning [9] that prohibited the sale and use of these devices after recording a high number of postoperative complications and adverse effects.

Nowadays, international guidelines [10,11] recommend starting with conservative treatments such as rehabilitative therapy or lifestyle modifications before moving on to surgical options, with the midurethral sling (transobturatory or retropubic sling) being the most safe and effective surgical treatment available. Recently, single-incision slings, another form of midurethral sling, were developed as a novel technology capable of treating SUI while ensuring reduced invasiveness and postoperative complications for patients.

Evolving towards treatment requiring less invasiveness, next to single-incision slings, bulking agents represent, nowadays, an alternative option for patients with comorbidities and those who are less suitable for the usual surgical treatment due to the chance of having to repeat the procedure frequently and risk a higher tax of recurrences.

SIS (single-incision slings) and UBA (urethral bulking agents) are increasingly being used as first-line treatment options to try to reduce complications.

SISs are known to result in less post-operative groin pain, less bleeding, and shorter surgical times than traditional slings. On the other hand, UBAs are considered less effective in the long term but show fewer total complications.

UBA therapies were traditionally used to treat women with painful SUIs caused by ISD [12]. Currently, UBAs are rarely used as a first-line therapy for SUI; however, this procedure might be preferred by women who would rather have fewer postoperative problems, in lieu of performing this treatment several times to reduce the likelihood of a SUI recurrence.

In order to enhance the minimally invasive options available to patients who, in today’s world, require effective and noninvasive treatments, this study compares UBAs and SISs in terms of efficacy, quality of life, and sexual function in women receiving first-line treatment for stress urinary incontinence. It also aims to demonstrate that UBA could be a viable first-line alternative to more invasive treatments.

## 2. Materials and Methods

From January 2016 to January 2021, 159 consecutive patients affected by SUI were included in the study. A prospective observational analysis was performed.

All data were prospectively evaluated from a urogynecological internal database. The Institutional Review Boards (IRB) approved the study (protocol number CD-27/2016). Informed written consent was obtained from all women. The research was conducted according to Good Clinical Practice Guidelines. Sixty-four patients underwent treatment with a UBA and 75 were treated with an SIS. As a multicentric prospective study, the clinical investigation was conducted at Sandro Pertini Hospital (Rome, Italy), Institute Marco Pasquali ICOT (Latina, Rome, Italy), Pia Fondazione G. Panico Hospital (Tricase, Italy), Policlinic Campus Biomedico Hospital (Rome), and Santo Spirito Hospital (Rome, Italy). Physical examinations, voiding diaries, and urodynamic tests were performed at our urogynecology outpatient clinic at the beginning and at the end of treatment.

The present study included only individuals who had symptoms for more than 1 year, had failed conservative treatment, and had incontinence episodes more than once every 24 h.

The following conditions were used as exclusion criteria: pelvic organ prolapse superior to grade 2(POP-Q system),neurogenic bladder, pure UUI (urgency urinary incontinence) and/or exclusive symptoms of OAB(overactive bladder), ongoing and/or suspected breast cancer, ongoing and/or suspected hormone-dependent tumors, urological tumors, endometrial hyperplasia and atypical uterine bleeding, ongoing or past venous thromboembolism, clinical evidence of chronic inflammation or urinary tract infection, and treatment history involving pelvic radiation.

Each patient conducted a supine and standing cough stress test at 300-mL bladder filling during the urogynecological examination. Urodynamic examinations were carried out in accordance with the International Continence Society (ICS) guidelines.

The maximum urethral closure pressure of 20 cm H_2_O and the Valsalva leaking point pressure of 60 cm H_2_O were regarded as indicators of intrinsic sphincter deficiency.

All patients in this investigation exhibited urodynamically verified urinary stress incontinence, with a median maximum cystometric capacity of 322 mL (ranging 245–498 mL) and a median Valsalva leaking pressure of 59 cm H_2_O (ranging 40–100 cm H_2_O), and no indication of outflow obstruction (Qmax 15 mL/s, Pvesmax > 50 cm H_2_O). Patients with concurrent urinary tract infection, previous surgery for stress incontinence, functional bladder capacity of 200 mL, and stage 2 pelvic organ prolapse were excluded from the study.

Moreover, the patients completed a voiding diary before and after the treatment. Postoperatively, the International Consultation on Incontinence Questionnaire-Urine Incontinence-Short Form (ICIQ-UI-SF) was completed to assess the impact of urine symptoms. To assess sexual function, the standardized Female Sexual Function Index (FSFI), the Female Sexual Distress Scale (FSDS) and the Pelvic Organ Prolapse/Urinary Incontinence Sexual Questionnaires short form (PISQ-12) questionnaires were administered on the first visit and again after 3 months. Finally, after treatment, the Patient Global Index of Improvement (PGI-I) was calculated.

Cefazoline 2 g was administered to all patients as a preventative measure 30 min before surgery.

The patient was placed on the operation table with her hips slightly flexed.

In UBA group a local anesthetic-containing lubricant was applied within the urethra, followed by a gradual trans-urethral instillation of 2% lidocaine solution. The PAHG injection was performed under endoscopic control with a single-use PAHG Bulkamid^®^ cystoscope (Contura International A/S, Sydmarken 23, 2860 Soeborg, Denmark) linked to a 0-degree optic to provide precise and accurate PAHG submucosal injection. The rotating sheath over the cystoscope allows the working channel to revolve 360-degrees, allowing for better access and visual control of the injection sites without having to move the entire cystoscope. Technique points include cautious needle advancement to avoid unintentional urethral mucosa injury and an angulation of fewer than 5-degrees to avoid too-deep injections. The best submucosal injection locations are at 2, 5, 7, 10 a.m., and 1 cm within of the bladder neck (proximal urethra). To ensure good urethral wall coaptation, 1–2 mL of Bulkamid^®^ (Contura International A/S, Sydmarken 23, 2860 Soeborg, Denmark) are injected at four sites, with no more than 0.5 mL injected at each site.

In the SIS group, a spinal anesthesia was performed. The Altis^®^ Single Incision Sling System is a transobturator MUS (midurethral sling) that is adjustable and authorized for the treatment of stress urine incontinence.

Approximately 1 cm proximal to the urethral meatus and extending downward towards the bladder neck, a 1.5 cm midurethral incision was made on the anterior vaginal wall. Then, the scissors are inserted into the vaginal incision and a “push spread” technique (at least 1.5 cm wide) is used to dissect back to the ipsilateral ischiopubic ramus. Secondly, the introducer and sling are inserted into the midline vaginal incision using an inside-out approach, with the tip of the introducer targeted via the previously dissected periurethral site towards the obturator membrane landmarks (a “10” and “2” o’clock locations). Finally, the sling is adjusted by dragging the suture loop across the patient’s midline until the required support is attained, and it should be positioned tension-free beneath the urethra, allowing a right-angle tool to easily slip between the sling and the urethra. This sling employs one static and one dynamic anchor at either end of a pulley suture, allowing for simple intraoperative tension modulation.

Clinical evaluation and exam, uroflowmetry, urodynamic exam and questionnaires were performed at the first appointment and after at least 24 months after surgical intervention.

Using Fisher’s exact test, we determined the statistical significance of each event based on its incidence. For each comparison, an odds ratio (OR) and 95% confidence interval (CI) were generated. To evaluate whether data were sampled from a Gaussian distribution, normality tests (D’Agostino and Pearson tests) were used. To compare continuous parametric and non-parametric variables (data that do not fall into a normal distribution), the *t*-test and Mann–Whitney U test were employed, respectively. The Spearman rank coefficient was used to calculate correlations between numerical parameters. A matched *t*-test was used to evaluate the change in questionnaire results (ICIQ-UI-SF, PISQ-12, FSFI, FSDS, PGI-I). All analyses were carried out with the Statistical Package for the Social Sciences (SPSS) 22.0 for Mac (SSPS, Chicago, IL, USA). A *p*-value of less than 0.05 was considered significant.

## 3. Results

The total number of patients was 211. Since 33 patients did not match the inclusion criteria and 39 were lost to follow-up, the sample consisted of 159 patients.

The 159 evaluated patients were divided into 2 groups: 64 patients who underwent UBA and 75 with SIS. Patients’ characteristics as age, BMI, parity, menopausal status, use of HRT, previous hysterectomy and POPQ status are similar between the two groups and shown in Table 1.

Median follow-up was 29 months (24–37).

The two procedures had an almost overlapping intervention time (22.87 + 6.32 min. for the bulking-agents injection vs. 23.22 + 7.44 for ALTIS). We registered the post-operative complications in both groups (Table 2) but only two of them reached a statistical significance; no patients of the Bulkamid group complained about pain after the procedure, unlike the ALTIS group where 9 patients out of 75 (10.8%) experienced post-operative groin pain (*p* = 0.03). Five patients in the ALTIS group developed de novo urgency compared to none in the UBA group (0.04).

The comparison of the voiding diary before and 29 months after treatment also showed interesting results (Table 3).

In the Bulkamid group, only 4 patients (5.1%) had a persistently positive Stress Test (*p* < 0.0001), as well as the ALTIS group in which 5 patients (6.2%) had a positive Stress Test (*p* < 0.0001).

The results of the urodynamic assessment conducted both before and after the therapy are shown in Table 4. In terms of first voiding desire (from 91 to 138 mL in the Bulkamid group and from 89 to 142 mL in the ALTIS group), maximal cystometric capacity (from 301 to 387 mL in the Bulkamid group and from 298 to 398 mL in the ALTIS group), detrusorial pressure at peak flow (from 18 to 14 cm H_2_O in the Bulkamid group and from 19 to 13 cm H_2_O in the ALTIS group), and peak flow (from 20 to 23 mL/s in the Bulkamid group and from 19 to 25 mL/s in the ALTIS group), the table demonstrates the significant outcomes achieved with these treatments and supports the efficacy of both methods without differences between the 2 groups.

In both groups, we observed a notable improvement in the QoL (quality of life) with a halving score in ICIQ-UI-SF 29 months after treatment (Bulkamid group from 14.58 ± 5.11 at baseline to 5.67 ± 1.90 after 29 months; *p* < 0.0001 vs. ALTIS group from 13.75 ± 5.89 to 5.83 ± 1.78; *p* < 0.0001).

Likewise, we noted an improvement in sexual function, with the number of sexually active patients increasing from 29 to 44 (56.4%) in Bulkamid group (*p* = 0.041) and from 31 to 51 (61.7%) in ALTIS group (*p* = 0.034). In accordance with the last data, the scores derived from PISQ-12, FSFI and FSDS also showed an improvement in women’s sexual function (Table 5).

## 4. Discussion

Our analysis compares the two primary minimally invasive treatment choices for SUI above a wide range of additional possibilities for the first time in the literature: the contemporary SIS vs. UBAs. The comparison is based on the assessment of treatment effectiveness, safety, and enhancement of sexual function and quality of life.

SUI is a frequent condition among women and has a significant impact on quality of life (QoL). The first-line approach should include conservative therapies such as lifestyle advice, physical therapies (PFMT and pelvic-floor muscle training), scheduled voiding regimes, behavioral therapies, and medications [5]. When all of these therapies fail, patients with limited bladder-neck mobility may undergo the full range of surgical treatments, such as midurethral sling, gold standard treatment, or, when indicated, UBA injection.

Since retropubic and transobturator midurethral slings are associated with severe adverse effects (including bladder rupture, damage to blood vessels, and pelvic pain), today, single-incision midurethral slings (SISs) aim to reduce complications and be less invasive.

Nowadays, the literature acknowledges that SIS is an excellent and effective technique despite being minimally invasive, with significantly reduced operating times and pelvic inguinal pain compared to traditional approaches [13].

Contrary to UBAs, an SIS is also commonly used as a first-line treatment for SUI. UBAs are now largely indicated for IDS and/or urethral hypomobility. However, in both circumstances, they are considered a second-line alternative treatment.

Our initial goal was to compare UBAs and SISs in order to establish the efficacy of both these minimally invasive treatments.

Indeed, much has been published about women with SUI preferring less-successful interventions with fewer postoperative complications to more-effective procedures with significant side effects [14,15].

Regarding safety, only eight patients out of 64 from the UBA group showed complications. Only one patient experienced acute urinary retention, in contrast to Giammò et al. who described 8.2% of patients experiencing this self-limited side effect [16]. None of our patients reported de novo urgency, unlike Itkonen Freitas et al. [17] who described 9.3% of patients experiencing this complication. On the other hand, 26 patients out of 75 from SIS group showed side effects. The most frequent complication in this group was groin pain, in line with Moran et al. We also registered 6.6% of patients showing de novo urgency, which is comparable with the 5.3% of patients registered by Youxiang Han et al. [18], while Moran et al. reported a slightly higher rate of de novo urgency (8.1%) as well as a higher rate of urinary retention cases (7.2%) compared to our single case. Tape extrusion occurred in 2.5% of patients, in line with the literature [19,20].

SISs and UBAs were demonstrated to be highly effective. This is shown in the stress-test data. In fact, at the median follow-up (29 months), the number of patients with positive stress tests decreased drastically. Similarly, Q-Tip Swab Test grades were almost halved in both groups. These results appear to be better than the average “objective cure rate” drawn from the studies we analyzed [15,17,19,21].

We could justify this high cure rate because all procedures had been performed by the same expert (more than 100 procedures) surgeon, in the same center with a high volume of patients. Nevertheless, we believe in the need to standardize the parameters that define the “objective cure rate”, to align outcomes of these two procedures.

Another fundamental parameter to assess the effectiveness of treatments is the “subjective cure rate”, which could be defined as the personal perception of clinical improvement by the patients.

We obtained this data by submitting, to patients, questionnaires to evaluate their QoL, such as the ICIQ-UI-SF and the PGI-I scale. According to Kamarkar et al. [22], the cut-off points in the ICIQ to evaluate patients’ satisfaction should be <6/21. These results are supported by the data in the literature which show a notable improvement in the ISIQ-UI-SF score after treatment [17,22]. The other item we used to evaluate QoL was the PGI-I scale. In our study, patients reported to feel “very much better” or “much better”, so the “subjective cure rate” of both groups after treatment was approximately in line with the literature [15,17,19,21,23]. Hence, SISs and UBAs appeared to be totally comparable in effectiveness and safety at a 24-month follow-up. The only significant difference was the absence of groin pain after UBA treatment.

Another goal of our study was to investigate the changes in sexual function and sexual satisfaction of women treated with bulking agents vs. SIS. The number of sexually active patients (>2 intercourses/month) increased from 29 (37.2%) to 44 (56.4%) in the UBA group and from 31 (38.2%) to 50 (61.7%) in the SIS group. Similarly, the scores from the PISQ-12, FSFI, and FSDS showed an improvement in women’s sexual function. There is limited literature available on the evaluation of sexual life after surgical SUI treatment. The two studies we found assess sexual function by using only one questionnaire out of the three we used in our study [24,25].

The strength of our study lies in the mid-term follow-up, which enables the evaluation of patients over time, unlike studies with only a short-term follow-up. As mentioned above, all patients underwent treatment by a single surgeon in the same high-volume center, minimizing the inter-operator outcome variability. Evaluation of sexual function via more than one questionnaire allows for the creation of a more precise score for sexual activity.

Some limitations include the small number of patients, the need of a longer follow-up (>60 months), the presence of selection bias, and the absence of randomization. In addition, we found, in Sekiguchi et al. [26], a cumulative cure rate of 91% after SIS treatment in a group of patients affected by mixed urinary incontinence, showing SUI together with ISD characteristics. According to these results, various research [27,28,29] showed high rates of success and enhanced quality of life following SIS therapy. This could widen the field of the application of SIS treatment, but further studies and investigations are needed, including a randomized double-blind-design study on a larger cohort of patients.

## 5. Conclusions

Overall, to the best of our knowledge, this is the first comparative evaluation of these therapies in two groups of patients with comparable features. Furthermore, using a variety of tools for evaluation, our study assesses both the objective and subjective success of the therapies.

Although further research and double-randomized trials are required, we have demonstrated that UBAs are highly successful when compared to minimally invasive surgical methods such as SISs, and they also have fewer side effects. Our study shows how a UBA can be used as a first-line therapy option since it helps reestablish the transient sphincteric mechanism of continence, which is the foundation of incontinence physiology. This gives women the option to choose the therapy that makes them feel more comfortable and gives them the possibility to choose a less invasive procedure.

## Figures and Tables

**Table 1 healthcare-12-00751-t001:** Patient characteristics.

Variable	Bulkamid Group (64 Patients)	ALTIS Group (75 Patients)	*p*
**Age, year**	55.8 ± 10.2	56.8 ± 8.9	0.06
**Body mass index, kg/m^2^**	27.5 ± 3.4	26.8 ± 5.7	0.06
**Parity, range**	2 (1–3)	2 (1–3)	0.08
**Menopausal status, n (%)**	31 (48.4)	32 (42.7)	0.07
**Hormone replacement therapy, n (%)**	8/31 (25.8)	8/32 (25)	0.09
**Previous hysterectomy, n (%)**	9 (11.5)	7 (8.6)	0.08
**POPQ system**			
**Stage 0 (%)**	60 (76.9)	59 (72.8)	0.08
**Stage 1 (%)**	18 (23)	22 (27.1)	0.07

Values are given as mean ± standard deviation (SD).

**Table 2 healthcare-12-00751-t002:** Postoperative complications in 159 patients.

Variable	Bulkamid Group (64 Patients)	ALTIS Group (75 Patients)	*p*
**Operative time, min**	22.87 ± 6.32	23.22 ± 7.44	0.08
**Fever, n (%)**	1 (1.2)	0 (0)	0.09
**Groin pain, n (%)**	0 (0)	9 (10.8)	0.03
**Urinary tract infection, n (%)**	2 (2.6)	3 (3.6)	0.07
**Deep vein thrombosis, n (%)**	0 (0)	0 (0)	0.07
**Urinary retention for up to 7 days, n (%)**	1 (1.2)	1 (1.3)	0.08
**Tape extrusion, n (%)**	0 (0)	2 (2.5)	0.08
**Severe pain, n (%)**	0	0 (0)	0.09
**Dyspareunia, n (%) †**	0	2 (2.5)	0.06
**De novo urgency (%)**	0	5 (6.6)	0.04
**Recurrent SUI (%)**	4 (5.1)	4 (5)	0.09

SUI = Stress urinary incontinence. †: in patients who regularly practice sexual activity (>2 intercourses/month).

**Table 3 healthcare-12-00751-t003:** Comparison of Voiding Diary before and after treatment (29 months Follow-Up).

Variables	Bulkamid Group (64 Patients)			ALTIS Group (75 Patients)			
Follow Up	Baseline	Median FU	*p*	Baseline	Median FU	*p*	*p*
**Positive Stress Test (%)**	64 (100)	4 (5.1)	<0.0001	75 (100)	5 (6.2)	<0.0001	0.07
**Q-Tip swab test (grade)**	41.44 ± 12.10	23.15 ± 10.41	<0.0001	42.34 ± 11.11	21.87 ± 8.56	<0.0001	0.08
**Mean number of voids (24 h)**	7.72 ± 1.65	9.43 ± 2.22	0.04	7.34 ± 2.12	7.65 ± 1.98	0.08	0.03
**Mean number of nocturia events**	0.98 ± 0.43	0.78 ± 0.45	ns	1.12 ± 0.88	0.92 ± 0.95	0.06	0.09

Abbreviation: ns: not significant.

**Table 4 healthcare-12-00751-t004:** Pre- and post-urodynamic evaluation.

Urodynamic Data	Bulkamid Group (64 Patients)			ALTIS Group (75 Patients)			Bulkamid vs. Altis
	Baseline	12 Weeks	*p*	Baseline	12 Weeks	*p*	*p*
**Peak flow (mL/s)**	20.71 ± 3.60	23.23 ± 4.23	0.01	19.65 ± 4.23	24.81 ± 5.88	<0.0001	0.07
**Flow time (mL/s)**	26.22 ± 5.11	27.67 ± 5.18	0.11	25.68 ± 5.51	27.77 ± 5.11	0.09	0.81
**Post-void residual (mL)**	20.55 ± 6.28	19.54 ± 6.12	0.49	21.11 ± 7.09	20.13 ± 7.11	0.54	0.72
**First voiding desire (mL)**	91.76 ± 20.13	138.72 ± 19.24	0.004	89.23 ± 21.47	142.43 ± 19.98	<0.0001	0.32
**Maximum cystometric capacity (mL)**	301.31 ± 73.56	387.76 ± 82.44	0.002	298.65 ± 77.28	398.26 ± 91.21	0.0031	0.55
**Detrusor pressure at peak flow (cm H_2_O)**	18.78 ± 5.63	14.45 ± 6.10	0.0012	19.11 ± 6.12	13.89 ± 4.89	<0.0001	0.21
**Maximum Urethral Closure Pressure (cm H_2_O)**	69.87 ± 9.11	70.32 ± 8.34	0.69	68.91 ± 9.71	71.09 ± 7.91	0.51	0.72
**Urethral Functional Length (mm)**	28.10 ± 2.22	28.21 ± 2.33	0.41	28.43 ± 3.01	28.67 ± 2.93	0.65	0.81
**Patients with detrusor overactive (%)**	36 (60)	23 (38.3)	0.13	30 (57.7)	9 (17.3)	0.02	0.08

**Table 5 healthcare-12-00751-t005:** Quality of Life and Sexual Function at 29 months follow up.

Variables	Bulkamid Group (64 Patients)			ALTIS Group (75 Patients)		
	Preoperative	Median FU	*p* Value	Preoperative	Median FU	*p* Value
**ICIQ-UI-SF**	14.58 ± 5.11	5.67 ± 1.90	<0.001	13.75 ± 5.89	5.83 ± 1.78	<0.001
**Sexual Activity † (%)**	29 (37.2)	44 (56.4)	0.041	31 (38.2)	50 (61.7)	0.034
**PISQ-12 ‡**	30.44 ± 7.23	36.54 ± 6.98	<0.001	31.22 ± 5.65	38.33 ± 6.24	<0.001
**FSFI ‡**	20.43 ± 2.22	29.77 ± 1.89	<0.001	21.21 ± 1.43	29.34 ± 2.11	<0.001
**FSDS ‡**	21.65 ± 4.76	8.32 ± 3.56	<0.001	20.98 ± 5.43	7.86 ± 4.78	<0.001

Abbreviations: ICIQ-UI-SF: International Consultation on Incontinence Questionnaire–Urinary Incontinence Short Form; PISQ-12: Pelvic Organ Prolapse/Urinary Incontinence Sexual Questionnaire short form; FSFI: Female Sexual Function Index; FSDS: Female Sexual Distress Scale. †: Number of patients who regularly practice sexual activity (>2 intercourses/month). ‡: In patients who regularly practice sexual activity (>2 intercourses/month).

## Data Availability

Data are contained within the article.

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
