# Peer review of "Minimally Invasive Treatment of Stress Urinary Incontinence in Women: A Prospective Comparative Analysis between Bulking Agent and Single-Incision Sling"

_healthcare, 2024, doi:10.3390/healthcare12070751_

Round 1
Reviewer 1 Report
Comments and Suggestions for Authors
First of all, I want to congratulate the authors for their manuscript “Ultraminimally invasive surgery in stress urinary incontinence treatment: prospective comparative analysis between bulking agent and single incision sling”.
In their study, the authors compare two completely different surgical approaches in the treatment of stress incontinence in women. Therefore, they prospectively included 159 women, from which 64 were treated with bulking agents and 75 patients received a single incision sling procedure.
Aims of interest had been the comparison of both procedures in regard of treatment success, patients´ quality of life after treatment and occurrence of complications because of the treatment.
This is a very interesting topic and since urinary stress incontinence is a bothersome burden for patients with a high impact on healthcare costs, it is important to endorse studies dealing with treatment strategies for patients benefit.
But I am afraid, that the manuscript - as it is presented by the authors for now - has major flaws, which make an acceptance impossible.
Overall, the whole manuscript does not seem to be written at leisure because of many mistakes in punctation marks, abbreviations, etc. A conclusion is missing, the abstract is not subdivided, etc.
Title:
- The Title is not clear. You treated women and this should appear in the title. As well as what kind of urinary stress incontinence (see following points).
Abstract:
- The Abstract would have more of a scientific character if it had subheadings, like “Aims and objective, Material and Methods,…” and it would be easier to follow.
- Results should be displayed by numbers. “More efficacy and safety by bulking agents”… safety in regard of the surgery or in the outcome of the incontinence? The whole abstract seems to be very loosely expressed.
- Abstract ends with describing of results: “….Furthermore,….. .....and those treated with single-incision slings.” (PS: Punctuation mark missing). What is your conclusion? What does your own study and your own results tell you? No interpretation of your own results. By now you lost most of your readers.
Introduction:
- Overall, the introduction is very confusing and in my opinion not to the point. You describe most of the time etiology and different theories of stress urinary incontinence. This should be mentioned in a short abstract. But the main topic of your study are the different treatment strategies. You should provide more information about buking agents and single incision slings and - above all - the indication for those surgeries.
- You do not mention other - well established - treatments, that are primarily used for severe incontinence and pelvic floor weakness, like sacrocolpopexie etc. - still gold standard. Slings and BA are not suitable for every indication and all kinds of severeness and should be clearly mentioned.
- What are contraindications for BA and SIS?
- Please provide references. E.g. line 52-54.
- The different paragraphs seem not to be in order. E.g. line 61 to 62.
- Which warning by the FDA? Line 61. Provide more information and references!
- The abbreviations seem to be out of order! (some are never written, some are when mentioned for the third time, some are mentioned more than ones). This happens throughout the whole manuscript.
- The introduction does not provide the understanding why this study is valuable.
Material and Methods:
- Ongoing problem with the abbreviations, what makes reading and understanding very difficult (e.g. line 88 (UUI)).
- Diagnostics before surgery: functional examination is very well done and described, the exclusion criteria are correct and the questionnaires are as well, but where are the diagnostics about the etiology of SUI? Do you perform perineal or transvaginal ultrasound? Or MRI? What about cystocele, rectocele? Those are very important facts in my opinion, which lead to the right choice for the right treatment.
- Could you provide information about the scoring - and evaluation systems of the questionnaires? Provide References!
Results:
- Table titles should be more precise.
- “Not significant” in the tables should be replaced by actual numbers, to be able able to see possible trends.
- Line 177-178: should not be part of the results.
- Commonly used classification of adverse events or complications is the Clavien Dindo classification. Could you provide this information as well?
Discussion:
- Line 216: As an introduction for the discussion this sentence could give the wrong impression, that those are the main – and only – surgical treatment options in SUI in women. Please rephrase (s. above).
- Abbreviations. E.g. Line 221 “PFMT”, Line 235 ……
- As mentioned above: Discussion about other surgical therapy options and how the ones you present fit in this “armada”.
- Lack of references (E.g. paragraph 225-8)
- Line 235-236: Key sentence for the introduction of the discussion, but I am afraid that the sentence - as it is written - does not make sense. Please rephrase.
- Paragraph 240-243 is redundant, maybe delete?
- Line 268: should not be in the discussion
- In general you should discuss more your own results and place them into discussion with current literature.
- You should discuss more the possible follow up after 24 months. Is it possible, that BA lose their efficacy over time?
- Paragraph Limitations and Strength is well written.
Conclusion:
- Missing
Thank you very much for giving me the chance to read and review this manuscript.
Comments on the Quality of English Language
can be improved
Author Response
Dear reviewer,
Thanks for your revision. Here the point by point response:
- We updated the title.
- We fixed the abstract based on your ideas.
- We gave sources and explanations for the abbreviation. We attempted to clarify the notion of the introduction, even if we have to respond you that the therapies you highlighted (example: Sacrocolpopexy) are mostly for POP and not for SUI
- We described the abbreviations. No MRI or ultrasound were conducted; nevertheless, POP was evaluated and included in the study's exclusion criteria.
- We were able to correct the errors when feasible. Clavien dindo classification gave each patient a maximum value of 2.
- We explained the abbreviations and removed the superfluous bits. We included a concluding section.
Thanking you in advance for your kindness and cooperation, we are available for any explanation you may request and we wish you a pleasant day.
Kind regards,
Lorenzo Campanella MD
Reviewer 2 Report
Comments and Suggestions for Authors
Dear Authors,
I congratulate with your work. The topic is interesting. The paper is well written and easy to be read. There are, however, some issues that may require clarification.
Title:
- I would be more cautious with the use of the term "ultraminimally", I would rather address UBA and SIS as "minimally" invasive surgery
Abstract:
- The type of bulking agent should be specified as the available materials are not all the same
- I would state that you address the results in terms of subjective effectiveness
Introduction:
- The "urethro-centric" hypotesis is not new, it has been developed more than 15 years ago. I would suggest to report some references about it.
Materials and Methods:
- Did you also include mixed urinary incontinence?
- The bulking injection technique and the SIS implant technique are well described in literature and could be avoided in the text
Results:
- How did you randomize the patients? If you compare the results of two separate cohorts, you introduce a severe selection bias that make the comparison inconclusive
- How were patients selected?
- I would expect a lower median intervention time from UBA: how do you explain that?
- In table 2 one patient in the UBA group had a "tape extrusion": how can a tape-less procedure have such a complication?
- Do you mean "De novo" SUI ore "Recurrent" SUI? They are different complications.
- Did you evaluate a pad test? pad use? Did you define dry patients? A simple stress test (standing or laying) may not reproduce the same event in everyday life that evokes a leak (running, jumping, walling...). Literature show a very different objective (lower) and subjective (higher) dry rate: how do you explain your results?
- A reduction in mean number of voids and nocturia events is useful for OAB, not for SUI.
Discussion:
- I do not agree that UBA is a second-line treatment, whereas SIS can be first line treatments: what evidence supports this conclusion?
- 29 months is still a short time follow-up
Comments on the Quality of English LanguageEnglish language is fine, only minor revision required
Author Response
Dear reviewer,
Thanks for your revision. Here the point by point response:
- we agreed with your suggestion and we modified the title.
- we specified the type of bulking agent we used in the MS and we also added it to the abstract section.
- The results were evaluated using both subjective and objective criteria, such as the questionnaires, and both are presented in the paper to provide the reader with a comprehensive understanding of our clinical research.
- We added reference 7 and 8.
- No, they were all patients suffering from SUI.
- Since these are unusual practices, we would prefer to preserve them in the paragraph if at all possible, since they could appeal to the article's readers.
- This is a prospective observational study, with no randomization. This is now included in the limitations(discussion section).
- Inclusion and exclusion criteria are already present in the text and they were used for an accurate selection of the patients
- the little discrepancy in operating time is negligible and not significant
- we managed to fix the mistake in the table
- we managed to fix the mistake in the table
- we didn't evaluate pad test and we preferred not to insert data about it. Our evaluations were accurately based on a urodynamic analysis.
- It still represents that UBA have a favorable outcome
- I will cite the recommendations available in the 7th edition(2023) of the ICI-ICS manual "Bulking agents should not be offered as first-line therapy for those women desiring a “one-time” durable solution for primary or re- current SUI (Grade B). Bulking therapy is an option for selected individuals with SUI (e.g., poor candidates for anti-incontinence surgery or those de- siring an office-based, minimally invasive procedure) after appro- priate counseling regarding lack of long-term durability (Grade B). Bulking agents may be offered as therapy for recurrent or per- sistent SUI following anti- incontinence surgery, although these outcomes are likely inferior to repeat anti-incontinence surgery in the long term, particularly in the presence of Intrinsic Sphincteric Deficiency (Grade C)." This is the conclusion section of the article Dwyer PL, Karmakar D. Surgical management of urinary stress incontinence - Where are we now? Best Pract Res Clin Obstet Gynaecol. 2019 Jan;54:31-40. doi: 10.1016/j.bpobgyn.2018.10.003. Epub 2018 Oct 30. PMID: 30503362. "Despite mounting evidence that retropubic MUS are more effective in the long run and in women at high risk of failure with recurrent USI and ISD, the TOT continues to have high satisfaction and use across the world. This is likely related to avoiding the retropubic space, which poses a significant risk of visceral injuries and bleeding. Doctors who see a small number of patients frequently believe that TOT MUS and micro slings are more secure in their hands; hence, these slings with high success rates provide a significant alternative to retropubic slings. There is an ongoing need for further comparative studies with longer follow-up periods to examine the various surgical options for stress incontinence." Nowadays, TOT and SIS represent the first line treatment for SUI while UBA need to be further assessed as a possible first line treatment. Howewer, this article aims to be a starting point in order to consider UBA as a possible first-line treatment.
- This is the first study comparing these two methods; more research is obviously required, but in our view, the number of follow-up months is a weakness that was included as a study limitation.
Thanking you in advance for your kindness and cooperation, we are available for any explanation you may request and we wish you a pleasant day.
Kind regards,
Lorenzo Campanella MD
Reviewer 3 Report
Comments and Suggestions for Authors
This study represents a valid alternative to what is reported in the literature, however the long-term benefits of bulking agents seem limited [Bradley, G.C. Injectable Bulking Agents for Incontinence. 2018], indeed a retrospective study with a seven-year follow-up showed relief or reduction of incontinence in 67% of patients, in 11.1% there was no change, and 2.3% worsened. 19.5% needed additional surgical treatment for persistent SUI [Brosche, T.; Kuhn, A.; Lobodasch, K.; Sokol, E.R. Seven-year efficacy and safety outcomes of Bulkamid for the treatment of stress urinary incontinence. Neurourol. Urodyn. 2021, 40, 502–508].
Therefore the results of this study are limited; randomized clinical trail are needed.
Major points:
- line 60: reference
- lines 62-63: use of the abbreviation SIS and UBA without terminology
- line 78: IRB protocol number is missing
- line 81: eliminate the frase “the study was approved by the institutional review board”
- line 87: pelvic organ prolapse is considered as exclusion criteria?
- modify or delete the table 6
- explain phrase in line 237-239
- lines 259-260: what do you mean by expert operator? How many procedures should he have performed?
- in my opinion the expertize of surgeon could lead to limited results as reported in the bulking agents literature. Could you report this section into discussion
- in which hospital were the patients treated? is not reported
Author Response
Dear reviewer,
Thanks for your revision. Here the point by point response:
- reference added
- terminology explained
- protocol number added
- frase deleted
- We included pelvic organ prolapse superior to grade 2 as an exclusion factor.
- We deleted the table.
- Patients may prefer less intrusive methods (such as bulking agents) even if they are less successful or liable to recurrence in comparison to more effective procedures(single-incision slings) that are more susceptible to serious adverse effects.
- an expert operator is someone who performed more than 100 procedures
- The surgeon's experience might undoubtedly be a constraint; nevertheless, in our instance, the operations were performed by qualified surgeons who were able to finish the surgeries with minimal rates of postoperative complications.
- We performed the study at a number of hospitals because it was a multi-center study. We added them to the MS.
Thanking for your kindness and cooperation, we are available for any explanation you may request and we wish you a pleasant day.
Kind regards,
Lorenzo Campanella M.D
Reviewer 4 Report
Comments and Suggestions for Authors
Author Response
Dear reviewer,
we are delighted that our clinical study fulfilled all your request.
Regarding your minor concerns, we will answer you point by point:
- Given the word count restriction, we decided to focus on the study's main conclusions for the abstract part. Because of the limited writing space we had, we were really unable to describe all of the noteworthy results we acquired, including the acronyms from the surveys (explained in more detail in the paper). The effectiveness was evaluated using surgical complications, urodynamic data, and quality-of-life questionnaires. Finally, we sought to demonstrate that UBA, although not being considered first-line treatment today, may have the same efficacy as surgical treatments with fewer postoperative consequences, providing patients the opportunity to pick the best therapeutic option available (tailored medicine). The surgeon's expertise is included in the study's discussion and it was of course a decisive factor since surgical therapy is an operator-dependent technique. Considering that this is a prospective observational study, no randomization was done.
- Because no data was reported in the material and methods section, it is assumed that no topical estrogen was used by any of the participants during the trial. The injection sites were four, and we were able to fix it in the text. PGI was measured at the conclusion of the 29-month follow-up period, which served as the study's endpoint. For the statical component, we chose mentioning the MS to clarify our stance, as it is plainly specified that either Pearson or Spearman rank were used:
"To evaluate whether data were sampled from a Gaussian distribution, normality tests (D'Agostino and Pearson tests) were used. To compare continuous parametric and non-parametric variables (data that does not fall into a normal distribution), the T-test and Mann-Whitney U test were employed, respectively. The Spearman rank coefficient was used to calculate correlations between numerical parameters." Tape extrusion was recorder 3 times for both of the groups and it required reoperation.
- We were able to figure out and fix the incorrect percentages and statistics that we had unintentionally recorded. The Q tip swab test is mentioned in the discussion section.
- We clarified each of the acronyms you listed. ALTIS is not an acronym because it is the name of the brand of single incision sling that we used to carry out the investigation. FU stands for follow-up.
Looking forward to work again with you in the future, we wish you a pleasant day.
Kind regards,
Lorenzo Campanella MD
Round 2
Reviewer 1 Report
Comments and Suggestions for Authors
Dear Authors.
Thank you very much for updating your manuscript, which – in my opinion – is highly improved.
Title:
- Thank you for rephrasing the title.
Abstract:
- Thank you for dividing the abstract into subdivisions and providing results. The conclusion is clearly stated.
- Line 34-36: Decrease in ICIQ-UI-SF?
- Line 41: “….they reported….” Who reported? Patients? Bulking Agents?
Introduction:
- Line 62-64: I am afraid, but in my opinion this sentence does not make sense. rephrase?
- Line 64-65: “….. playing an important role in stress incontinence is …..” – do you mean “….stress incontinence surgery/therapy…”?
- The introduction is now much more to the point and provides a better understanding. Thank you
Materials and Methods:
- Line 119: what kind of conservative treatments had the patients received before inclusion for this study?
Results:
- Still not providing actual numbers instead of “n.s.”.
Discussion:
- Overall better Discussion
- Line 254: “….primary treatment choices for SUI ….”. As you mentioned in your Mat&Met, there are exclusion criteria that lead to other treatment options for SUI. This sentence states that these surgeries are the primary surgical options for all forms of SUI.
Conclusion:
- You conclude, that UBA have fewer side effects compared to traditional surgical methods. However, you did not evaluate these treatment options to other traditional surgeries. Therefore, your conclusion is not legit. You could discuss this issue and compare literature with your results in explicit regard to side effects/complications. Still, interstudy comparison is not possible and this statement should not be made, even when discussed in the Discussion section.
Again, thank you for your effort.
Comments on the Quality of English LanguageQuality of English can be improved
Author Response
Dear reviewer,
Thanks for your revision. Here the point by point response:
-
the International Consultation on Incontinence Questionnaire - Urine Incontinence - Short Form (ICIQ-UI-SF)was completed to assess the impact of urine symptoms. We specified it in the abstract.
-
We refer to the patients treated with bulking agents. We specified it in the abstract
- We chose to omit the statement since it might have been difficult to reword and comprehend for the reviewers.
- Yes, we refer to the stress incontinence surgery/therapy
- The main conservative treatment used to treat SUI are lifestyle modifications (diet, exercise, no smoking, etc.) and pelvic floor muscle therapies (PFMT), which play an important role in the treatment of stress urinary incontinence and, sometimes, are enough to fully cure the symptomatology and guarantee the patient's complete recovery.
- We provided the numbers.
- The exclusion criteria we cited in Mat&Met refer only to conservative treatment and, in this phrase, we are referring to surgical options. I will cite the 7th International Consultation on Incontinence(2023) guidelines: "In comparison to other procedures, PS(pubovaginal slings) are as effective as MUS(miduretheral slings) in the short term,however operating time and OS are significantly shorter with MUS. Laparoscopic colposuspension can only be recommended for the surgical treatment of SUI in women by surgeons with appropriate training and expertise. UBA are associated with improvement rather than resolution of SUI, wan- ing effects over time, and a high rate of retreatment. They may have more benefit in women with poor urethral coaptation, such as those with a poorly mobile urethra, previous radiation, and vaginal/periurethral scarring. Their ease of use and minimal anaesthetic requirements should be balanced against potentially high costs, partly due to the frequent need for repeat injections.
Single-incision mini-slings (SIMS) were introduced as an attempt to decrease complications associated with retropubic and tran- sobturator slings. These procedures theoretically require minimal anesthesia and may be performed under local anesthestic. A Cochrane Review in 2017 (113) evaluated 31 trials (n = 3290) comparing single incision slings to retropubic MUS. The TVT Abbrevo was designed to evoke less groin pain. The subjective/ objective cure of 76.9% and 96% for the TVT Abbrevo. All groin pain resolved by 5 weeks."
After considering conservative therapy, it is clear that the most promising and evidence-based authorized mininvasive surgical treatments for SUI are represented by UBA and SIS (even tough UBA provide several limitations we cited in the manuscript). - We changed the sentence in the conclusion section and you will find it as: "Although further research and double-randomized trials are required, we have demonstrated that UBA are highly successful when compared to minivasive surgical methods such as SIS, and they also have fewer side effects". Our prospective observational study aims to evaluate two minimally invasive approaches in terms of postoperative complications, quality of life, method safety, and effectiveness.
Thanking you in advance for your kindness and cooperation, we are available for any explanation you may request and we wish you a pleasant day.
Kind regards,
Lorenzo Campanella MD
Reviewer 2 Report
Comments and Suggestions for Authors
Dear Authors,
thank you for having taken into consideration my comments, I appreciate it.
The design of the study (non randomized prospective study with not clear inclusion/selection criteria) allows to express the result on each single cohort, but not a comparison between the two techniques, as the selection bias severely affects the outcomes. Furthermore, the lack of a pad test represents and important drawback (a stress test cannot represent everyday life events that evoke an urinary leak).
I do not agree that these are unusual practices, these devices have been available for medical use for more than a decade, there is plenty of scientific papers illustrating the surgical technique: the description does not add much interest.
I completely agree with the statement about Incontinence guidelines, however these guidelines have a major drawback, including studies on different bulking agents that are not on the market anymore.
Author Response
Dear reviewer,
Thanks for your revision. Here the point by point response:
- We changed the sentence in the conclusion section and you will find it as: "Although further research and double-randomized trials are required, we have demonstrated that UBA are highly successful when compared to minivasive surgical methods such as SIS, and they also have fewer side effects". Our prospective observational study aims to evaluate two minimally invasive approaches in terms of postoperative complications, quality of life, method safety, and effectiveness and we specified that further trials are needed.
-
Regarding the pad test, I will cite the7th International Consultation on Incontinence guidelines:"A pad test is a good instrument for evaluating the severity of uri- nary incontinence. However, it is not a perfect “gold standard” for UI severity.The main limitation of the pad test, as a diagnostic tool for UI, is the value for the cut off of a positive test and the inability to distinguish between types of incontinence. This limitation is especially important when this test is used as an outcome measure, when surgical treatment of stress urinary incontinence is evaluated, as these women can suffer “de novo” urgency urinary incontinence. The committee recommends invasive UDS after failure of initial non-invasive management of SUI. (Evidence Level 1 guidelines). Grade of Recommendaition A." . Our primary analysis was done using urodynamic testing, which is considered the gold standard in the characterization and investigation of UI.
3. We would like to keep the description of the techniques in the manuscript.
4. I will cite you the 7th International Consultation on Incontinence guidelines regarding the bulking agents available today: "4.1. Available Agents :Due to manufacturing cessation (delayed skin reactions, arthralgia, (192)), bovine collagen (Contigen®) was obsolete at the time of ICI-5, not included in this chapter except for comparison with other agents. Limited data were published regarding stabilised hyaluronic acid/dextranomer (NASHA-dx; Zuidex®), but the latest Cochrane Review (192) noted that women treated with NASHA-dx had significantly higher rates of injection site complications com- pared to bovine collagen (16% vs. 0%, respectively; RR 37.78, 95% CI 2.34 to 610) and this product was withdrawn from the U.S. market (192) but is available for vesicoureteric reflux. Similarly, the 2017 Cochrane review noted that evidence of particle migration has stopped the clinical use of Polytetrafluoroethylene (Polytef paste).
a) Carbon coated zirconium beads (Durasphere®), available USA, Europe, Australia (Hoe 2021, 214)
b) Calcium hydroxyl apatite (Coaptite®), available in USA and Turkey
c) Polydimethylsiloxane elastomer (PDMS; Macroplastique®), UK, Europe, Australasia, USA
d) Polyacrylamide hydrogel (PAHG; Bulkamid®). UK, Europe, North America, Australasia
e) Urolastic (Urogyn) consists of vinyldimethyl terminated PDMS, tetrapropoxysilane cross-linking agent, platinum vinyltetramethyl siloxane complex catalyst, and titanium dioxide radiopacifying agent.They present a list of the UBA available on the market and ready for usage, emphasising how numerous materials and producers were prohibited after observing adverse responses in patients.
Thanking you in advance for your kindness and cooperation, we are available for any explanation you may request and we wish you a pleasant day.
Kind regards,
Lorenzo Campanella MD